# Contribution of Non-Rainfall Water Input to Surface Soil Moisture in a Tropical Dry Forest

Maria Simas Guerreiro [1,*], Eunice Maia de Andrade [2], Marcos Makeison Moreira de Sousa [3], José Bandeira Brasil [3], Jacques Carvalho Ribeiro Filho [3] and Helba Araújo de Queiroz Palácio [4]

1   FP-ENAS, FCT, Universidade Fernando Pessoa, Praça 9 Abril, 4249-004 Porto, Portugal
2   Departmento de Conservação de Solo e Água, Universidade Federal Rural do Semi-Árido, Rua Francisco Mota, 572, Mossoró 59625-900, Brazil; eandrade.ufc@gmail.com
3   Departmento de Engenharia Agrícola, Campus do Pici, Universidade Federal do Ceará, Fortaleza 60455-760, Brazil; makeison.moreira14@gmail.com (M.M.M.d.S.); josebandeira@alu.ufc.br (J.B.B.); jacquesfilho@alu.ufc.br (J.C.R.F.)
4   Instituto Federal de Educação, Ciência e Tecnologia do Ceará, Rodovia Iguatu-Várzea Alegre, km 5, Iguatu 63503-790, Brazil; helbaraujo@ifce.edu.br
*   Correspondence: mariajoao@ufp.edu.pt

**Abstract:** Non-rainfall water input to surface soil moisture is essential to ecosystems, especially in dry climates, where a water deficit may persist for several months. Quantifying the impact of water gains by soil moisture at night will help to understand vegetation dynamics in dry regions. The objective of this study was to evaluate the non-rainfall water contribution to soil moisture content at the soil surface and how it minimizes the water stress on plants with predominantly surface roots. The experiment was conducted in a low-latitude, semiarid environment with a dry tropical forest regenerating for 42 years. The soil moisture and soil temperature were measured at one-minute intervals from June 2019 to August 2019 using four capacitive humidity sensors and thermometers, installed at depths of 5 and 10 cm. the soil moisture increased significantly ($p < 0.05$) during the night at both depths from June to August, when there was no rainfall. There is a definite contribution of nightly gains to alleviate vegetation water stress during the dry months. These results show the importance of dew for water availability and for dry tropical forests species in the months of water deficit.

**Keywords:** semiarid regions; non-rainfall water input; water stress

## 1. Introduction

The dry areas of the globe account for approximately one third ($22.6 \times 10^6$ km$^2$) of the total surface area. Hyper-arid, arid, semi-arid and sub-humid conditions correspond to 41.3% of the land surface [1], and changes in climate due to increased greenhouse effects and land use changes may impose transitions from semi-arid to arid conditions [2,3]. Tropical dry forests (TDF) are present in the five continents, of which 54% is located in South America [4], accounting for approximately 40% of all tropical forests [4,5]. These forests are home to the poorest people in the world [6].

TDFs show average monthly temperatures above 18 °C, annual precipitation below the 1800 mm isohyet, and rainfall events concentrated within four to seven months [7,8]. Rainfall is the main water intake in TDFs. The hydrologic regime in these regions depends on the spatial and temporal concentration of rainfall, with it not being uncommon that 70% of the annual precipitation occurs in one month alone [9] or that 90% of the total annual precipitation is registered during the 3–4 months of the wet season [10]. These regions are characterized by long periods of water deficit [11], and vegetation adapts to the irregular rainfall and extended droughts with small-leaved, thorny trees with twisted

trunks, and succulents and therophitic herbs that efficiently respond to the minimal levels of precipitation [9].

The imminent growth of the rainfall deficit makes it necessary to use alternative means of obtaining water, especially in dry forest environments, where small amounts of water are fundamental to maintain vegetation. The main source of non-rainfall water input to the soil is fog, dew, the redistribution of condensation from leaves (dewfall from overstory and understory canopies), hydraulic lift from the deeper to the shallower soil layers and water vapor adsorption [12–17].

Dew is recognized as an important source of water for many arid and semi-arid ecosystems, due to its contributions to the daily, seasonal and annual water balance [18]. Even though the soil surface temperature does not always drop below the dew point, the soil moisture (SM) content may increase in the upper layers at night, alleviating vegetation water stress [12,15,16]. The main factors related to dew formation include surface temperature, absolute humidity, relative humidity and wind speed in the surface layer. Therefore, these conditions may vary according to the climatic conditions of each region.

Despite generating relatively small amounts of water, dew can be of great importance for local water balance in arid, semi-arid and hyper-arid environments, especially in regions with very low rainfall averages, as is the case of the Taklimakan desert in China (total annual precipitation < 200 mm) where the accumulation of dew represents up to 36% of the total precipitate (12.9 mm) [16]. Despite the importance of dew in regions with water deficit, its contribution to soil surface moisture in the Brazilian semi-arid region is still understudied because of the difficulty of measurement.

Most studies have highlighted the impact of dew on soil moisture in dry regions at or above the 23° latitude [12–15,19,20], but little research has been conducted in the dry intertropical region. Moreover, most of these observations focused on its effect on vegetation, rather than how, and to what extent, its distribution can influence the dynamics of soil surface moisture.

In the actual context of climate changes, it is important to assess more water sources in TDS as the main rainfall input is expected to decrease and, thus, their contribution to the system. Therefore, the present study characterizes the effect of dew on the variability of soil moisture content at the surface layers of a vertic soil in a dry tropical forest. The objective of this study was to evaluate the non-rainfall water input contribution to the soil moisture content at the soil surface in minimizing the water stress on plants with predominantly surface roots.

## 2. Materials and Methods

### 2.1. Study Area

The study was undertaken in a low-latitude, seasonally dry tropical forest in a semi-arid environment located in the municipality of Iguatu, CE, Brazil, at 6°23′47″ S and 39°15′29″ W (Figure 1). The climate of the region is a BSh′ (semi-arid hot), according to Köppen's classification. The average historical annual rainfall is 997 ± 300 mm, of which 89% are concentrated from December to May [10]. The average annual potential evaporation is 2113 mm year$^{-1}$, and the aridity index is 0.48 [11].

The soil of the area is classified as a typical carbonate ebanic vertisol [21]. It belongs to a sedimentary rock formation with limestone rock below 36 cm, with the presence of A, B and C horizons at 0–5 cm, 5–36 cm and above, respectively. The physical–chemical characteristics of the soil are presented in Table 1. The bulk density is 1.50 g/cm$^3$, and the particle density is 2.58 g/cm$^3$.

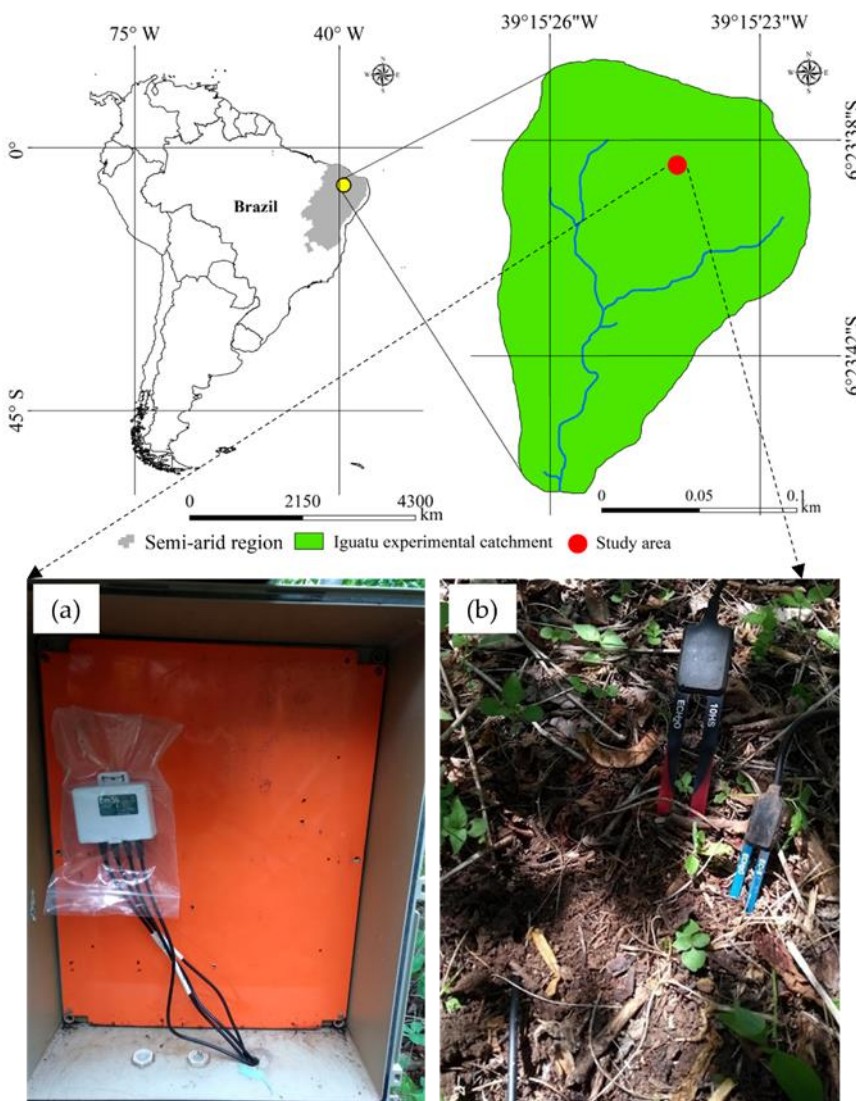

**Figure 1.** Location of the seasonally tropical dry forest fragment and experimental settings; (**a**) datalogger; (**b**) soil moisture sensors.

**Table 1.** Physical–chemical attributes of the soil of the study site. Determined by the authors, following standard procedures [22,23].

| Soil Physical/Chemical Parameters | | | |
|---|---|---|---|
| Attributes | Depth (10 cm) | Attributes | Depth (10 cm) |
| Clay (%) | 34 | H + Al (cmolc kg$^{-1}$) | 0.83 |
| Sand (%) | 24 | C (g kg$^{-1}$) | 17.76 |
| Silt (%) | 42 | $p$ (mg kg$^{-1}$) | 180 |
| Ca (cmolc/kg) | 38.8 | CE (ds m$^{-1}$) | 0.59 |
| Mg (cmolc/kg) | 4.5 | M.O (g kg$^{-1}$) | 30.62 |
| Na (cmolc/kg) | 0.17 | Ph (H$_2$O) | 7.3 |
| K (cmolc/kg) | 1.34 | Texture | Clay loam |

The land cover is a dry tropical forest under regeneration since 1978 after being used to grow crops. It is classified as closed-shrub arboreal dry tropical forest with deciduous trees, with predominance of the species Croton sonderianus, Mimosa caesalpiniifolia and

Aspidosperma pyrifolium [24]. The soil remains fully covered during the rainy season (Figure 2a), limiting the development of the herbaceous stratum, but the deciduous trees lost their leaves during the studied dry season (Figure 2b).

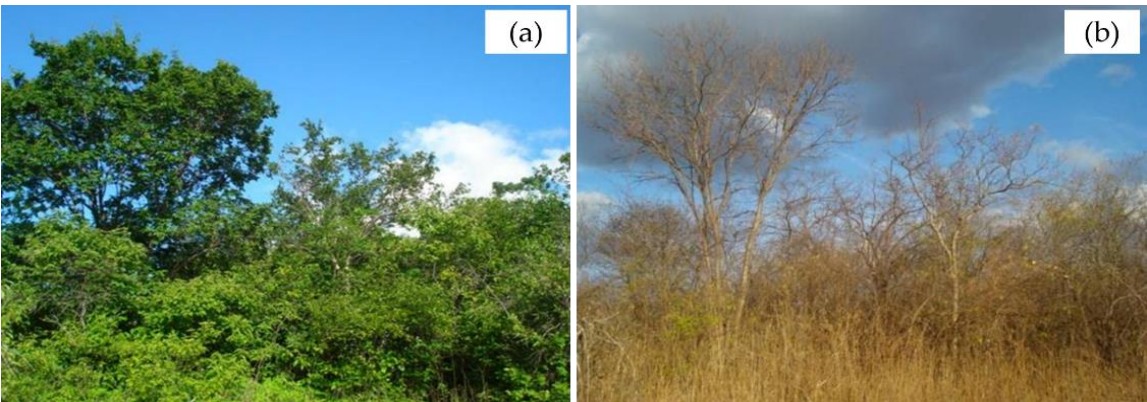

**Figure 2.** Tropical dry forest under regeneration: (**a**) wet season; (**b**) dry season. Photo by author.

*2.2. Meteorological Data*

This study was conducted from June to August, during the dry season, with no rainfall occurrences. The average number of daylight hours are 11:45, 11:47 and 11:54 for June, July and August, respectively. There was a small variation of 12 min for the sunrise hour—the sun rises at 5:42 a.m. on 1 June and 5:38 a.m. on 31 August, and the latest sunrise is on 18 July at 5:50 a.m. Based on the meteorological data from the Iguatu station (A319) at the IFCE Campus Iguatu [25], the radiation shows little variation over the studied period, with peak values varying from 987 to 1993 kJ/m$^2$ (Figure 3a), peaking at approximately noon (Figure 3b).

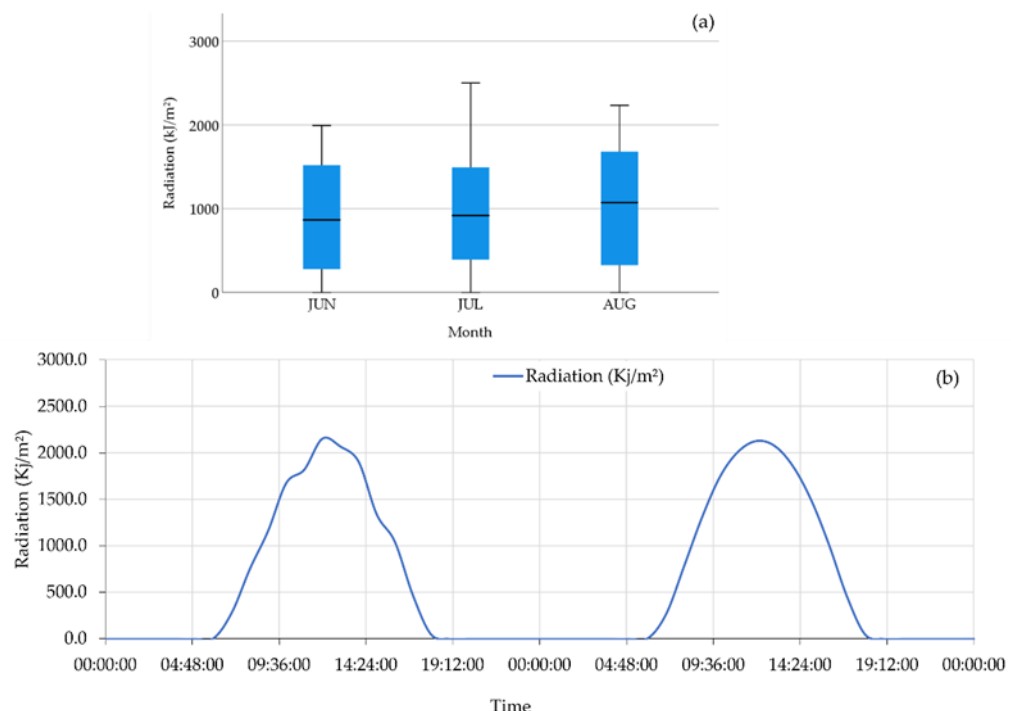

**Figure 3.** (**a**) Monthly variability of radiation June to August; (**b**) daily variation of radiation (23–24 August 2019).

The temperature and relative humidity vary in opposite directions, as expected. As the temperature increases, the relative humidity decreases (Figure 4a,b), with a minimum, max-

imum and average temperature in the period of 17.5 °C, 35.8 °C and 27.6 °C, respectively. The variability is greater during the day, and there is an increasing trend in temperature over the studied time period (Figure 4c).

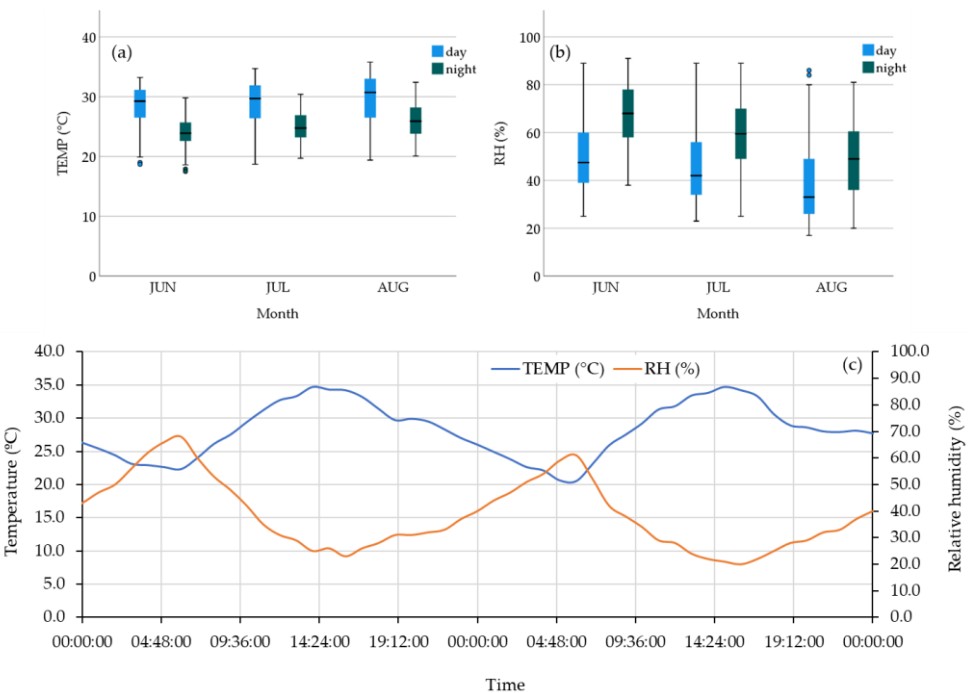

**Figure 4.** (**a**,**b**) Variability of temperature and relative humidity in the period of study; (**c**) daily variation of temperature and relative humidity (23–24 August 2019).

### 2.3. Soil Moisture and Soil Temperature Data

Continuous one-minute interval volumetric soil water content readings from EC-5 and 10HS sensors (www.metergroup.com (accessed on 4 March 2022)) and soil temperature from a Decagon RT-1 sensor (www.ictinternational.com (accessed on 4 March 2022)) were stored in an Em5b Decagon datalogger (Figure 1a) from January to December 2019. The equipment was installed near the existing weather station in the area. The sensors were placed at a depth of 5 cm and 10 cm (Figure 1b) to assess the soil moisture content for the 0 to 7.5 cm and 7.5 to 12.5 cm layers, respectively.

From the initially installed four soil moisture sensors, perpendicular to the soil surface and under the natural vegetation, two showed inconsistencies associated with high temperatures due to the electrical characteristics of the soil [26], and so, these sensors were discarded. To minimize the errors due to the high temperatures in the region, the remnant sensors were calibrated for gravimetric soil moisture content using the following equation:

$$\varpi = 12.7 + 0.756 \times \theta - 0.49 \times T$$

where $\varpi$ is the gravimetric soil moisture content ($g/g$) determined in the laboratory, $\theta$ is the volumetric soil moisture content obtained from the sensor ($m^3/m^3$) and $T$ is the soil temperature measured at a 10 cm depth (°C).

To analyze the contribution of non-rainfall water input to the soil–fog, dew and water vapor adsorption, we selected the data from the beginning of the dry season (June) until the sensors were no longer sensitive to soil moisture changes and records were unreliable when compared to the gravimetric method (at the end of August).

The soil moisture content and soil temperature data were not normally distributed ($p \leq 0.05$) by the Kolmogorov–Smirnov test. We applied the Wilcoxon test to verify the differences in medians from day to night soil moisture values ($p \leq 0.01$). For a descriptive analysis, the box plots were applied.

Daily SM gains and losses were calculated at both 5 cm and 10 cm depths, by the difference between the maximum and minimum SM values. The daily variation of those gains and losses was determined by the difference between the maximum and minimum SM in one day and the difference from the day before.

## 3. Results and Discussion

Being a low-latitude dry region, the water inputs to the soil are mostly from rainfall [27,28]—the soil moisture at 5 and 10 cm depths responds to rainfall, increasing after a rainfall event and decreasing between events (Figure 5), as expected [20]. The non-rainfall water inputs are fog, dew and water vapor adsorption [13,16]. During the study period, there were no occurrences of temperatures being below the dew point at any time; therefore, no dew was formed in the atmosphere (https://portal.inmet.gov.br/ (accessed on 14 July 2021)), with water adsorption being the primary pathway for the soil moisture content increase at night, as suggested by [12].

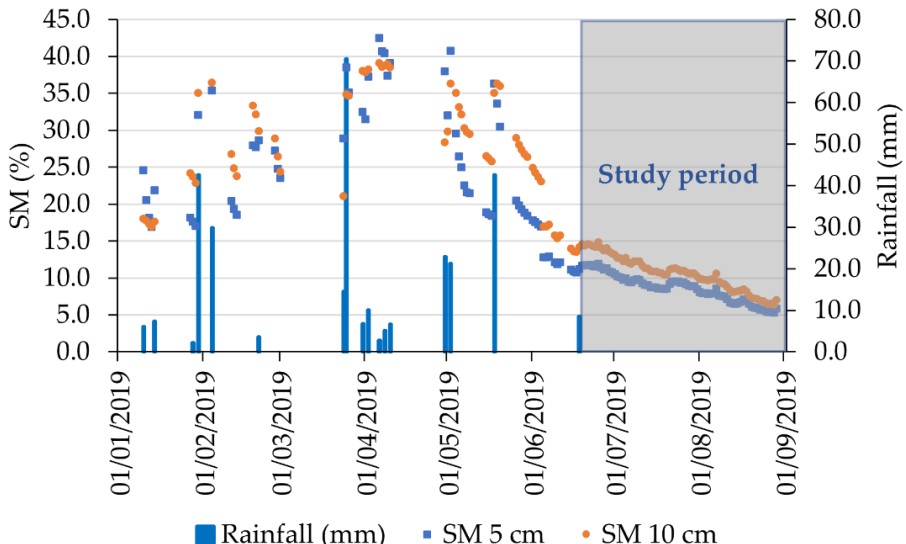

**Figure 5.** Average daily soil moisture (SM) content at 5 cm and 10 cm depths, and total rainfall.

As expected, the soil moisture content at the depth of 10 cm was higher than at 5 cm (89% of the time), especially during periods without rain (Figure 5), as the most superficial layer has greater interaction with the atmosphere, being subject to evaporation processes [29] and loss of water to the atmosphere. As this study is focused on the non-rainfall water input, we selected the records from the last rainfall event on 18 June to 31 August (Figure 5), because dew may occur up to three months after the end of the rainy season [30] and may persist throughout the dry season [31].

The highest average day and night soil moisture content was recorded at both depths (Figure 5, Table 2) in June, followed by July and August. Even though the soil moisture was at its lowest in August, the average soil moisture increased overnight (amplitude) at both depths of 5 and 10 cm. As the days were consecutively dry, there was a reduction in the soil moisture content, with some exceptions that are further discussed below.

As the soil temperature rises, adsorbed water in the soil evaporates to the soil pore spaces and is redistributed to the pore spaces. At the deeper layer (10 cm), part of the water vapor is transferred and redistributed to the upper layer (5 cm) by diffusion, of which some is lost to the atmosphere. The hydraulic lift process is not likely to occur with shallow root depths [17] as it does for deeper areas (60 cm), as shown by [32]. At night, as the soil temperature decreased, the water in the soil atmosphere condensed, and was adsorbed by the soil particles, increasing the soil moisture content. Similar results were reported by [16], where water vapor adsorption occurred when the relative humidity was lower

than the atmospheric relative humidity, even when the soil temperature did not reach the dew point.

**Table 2.** Average soil moisture values from the 44,735 readings performed during the study period.

| | Depth (cm) | SM (%) ± SD | | Min (%) | | Max (%) | | Amplitude (%) | | CV (%) | | Average Temp (°C) | |
|---|---|---|---|---|---|---|---|---|---|---|---|---|---|
| | | Day | Night | Day | Night | Day | Night | Day | Night | Day | Night | Day | Night |
| Jun | 5 | 11.31 ± 0.82 | 11.86 ± 0.71 | 10.1 | 10.5 | 13.41 | 13.39 | 3.31 | 2.89 | 7.2 | 6 | 26.6 | 25.3 |
| | 10 | 14.24 ± 1.39 | 15.21 ± 1.27 | 12.18 | 13.19 | 17.8 | 17.87 | 5.62 | 4.68 | 9.8 | 8.3 | | |
| Jul | 5 | 9.25 ± 0.80 | 9.84 ± 0.63 | 5.91 | 7.98 | 11.12 | 11.12 | 5.21 | 3.14 | 8.7 | 6.4 | 27.5 | 26 |
| | 10 | 11.35 ± 1.13 | 12.45 ± 0.82 | 8.37 | 9.85 | 13.95 | 13.98 | 5.58 | 4.13 | 9.9 | 6.6 | | |
| Aug | 5 | 6.00 ± 0.28 | 7.26 ± 0.87 | 2.45 | 4.98 | 8.84 | 8.85 | 6.39 | 3.87 | 21.4 | 12 | 30.5 | 27.2 |
| | 10 | 7.22 ± 1.66 | 8.97 ± 1.14 | 2.87 | 6.01 | 10.93 | 11 | 8.06 | 4.99 | 23 | 12.8 | | |

Despite the increased soil moisture content during the night, it was lost during the first hours of the day (Figure 6) when the soil temperature increased due to high-intensity solar radiation. Nonetheless, the increase in moisture in the first 10 cm of the soil can be a source of water for plants with absorption roots predominantly in the surface layer, thus decreasing the effects of the prolonged water stress in dry regions.

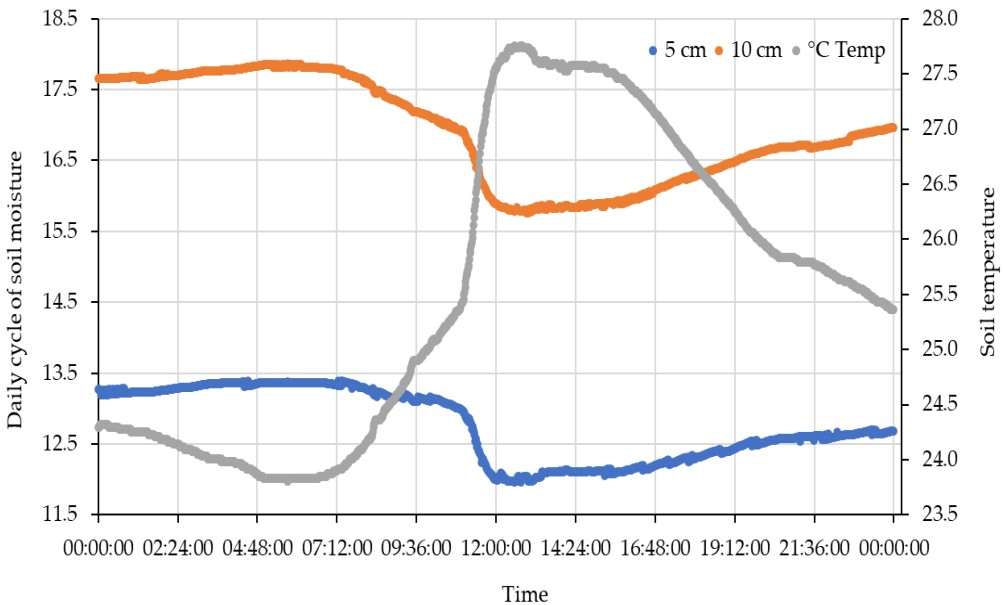

**Figure 6.** Daily cycle of soil moisture at 5 cm and 10 cm, and soil temperature.

The daily soil moisture gains during the night were lost during the day (Figure 7). Until 11 July, the night gains were lost to daily losses at the same rate, with no net gain or loss of stored water in the soil (Figure 7). As the dry season progressed, there was an increase in net losses with some exceptions. The radiation reduction observed on 21 July, 23 July, 5–7 August and 17 August impacted the cumulative losses in the same period, promoting gains rather than losses in those periods. This result shows the sensitivity of soil adsorption to meteorological factors, such as solar radiation—the soil moisture increase occurred when there was a reduction in the total daily radiation of over 20% from the previous day, and over 30% from the monthly average (Table 3).

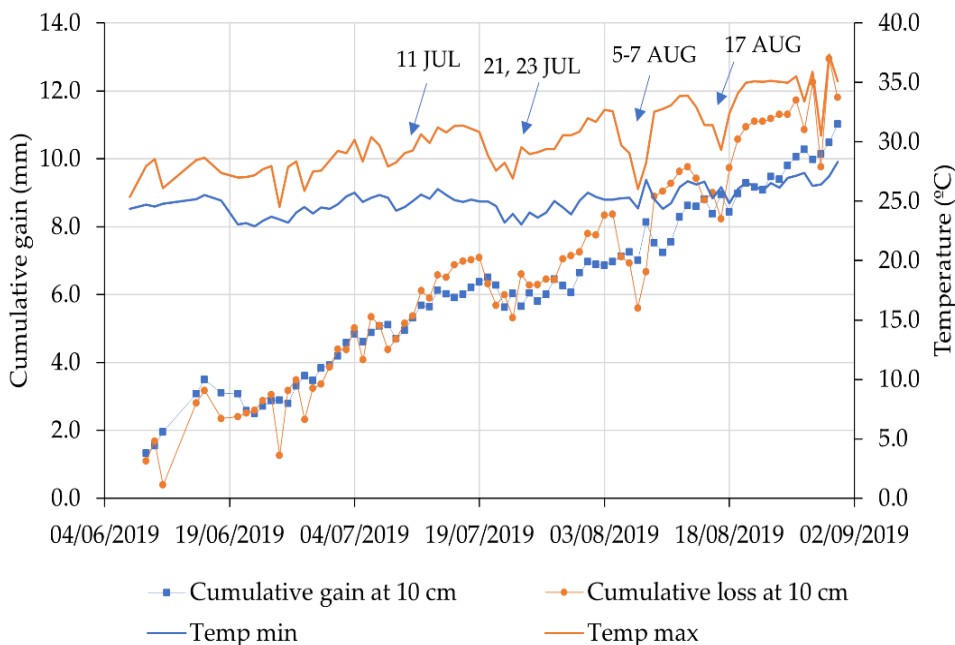

**Figure 7.** Time series of cumulative soil moisture gains and losses, and soil temperature.

**Table 3.** Total daily radiation on the day before and the day following soil moisture gains.

| Date | Total Daily Radiation (kJ/m²) | Date | Total Daily Radiation (kJ/m²) |
|---|---|---|---|
| 19 Jul | 13,548.38 | 4-Aug | 13,564 |
| 20 Jul | 9978.3 | 5-Aug | 7656.48 |
| 21 Jul | 7582.51 | 6-Aug | 7348.23 |
| 22 Jul | 11,904.18 | 7-Aug | 13,351.67 |
| 23 Jul | 7344.14 | 8-Aug | 13,138.74 |
| 24 Jul | 12,408.54 | 16-Aug | 12,270.43 |
| | | 17-Aug | 8779.38 |
| | | 18-Aug | 14,608.16 |
| | | 26-Aug | 14,419.86 |
| | | 27-Aug | 10,868.2 |
| | | 28-Aug | 15,675.22 |
| Mean radiation | 10,936 | Mean radiation | 13,442 |

The minimum soil temperature at night shows little variation, unlike the maximum soil temperature (Figure 8). This behavior shows the high sensitivity of soil moisture to solar radiation, even at the 10 cm depth in the period studied, which expresses the peak of higher losses. The soil moisture content never exceeded 13.4% and 17.9% at 5 cm and 10 cm depths, respectively (Figures 5 and 8a). The soil moisture was different during the day and at night ($p < 0.05$) in all months and at both depths of 5 and 10 cm. Even with the decrease in soil moisture averages over time, the soil continued to withhold water at night, recording increases in soil moisture at both depths in that period, as also observed by [12,13,15,16,18] in arid and semi-arid regions. The highest variability in the soil moisture occurred in August, when the soil temperature also had a higher variability both during the day and at night, and the temperature was at its highest value in this period (Figure 8).

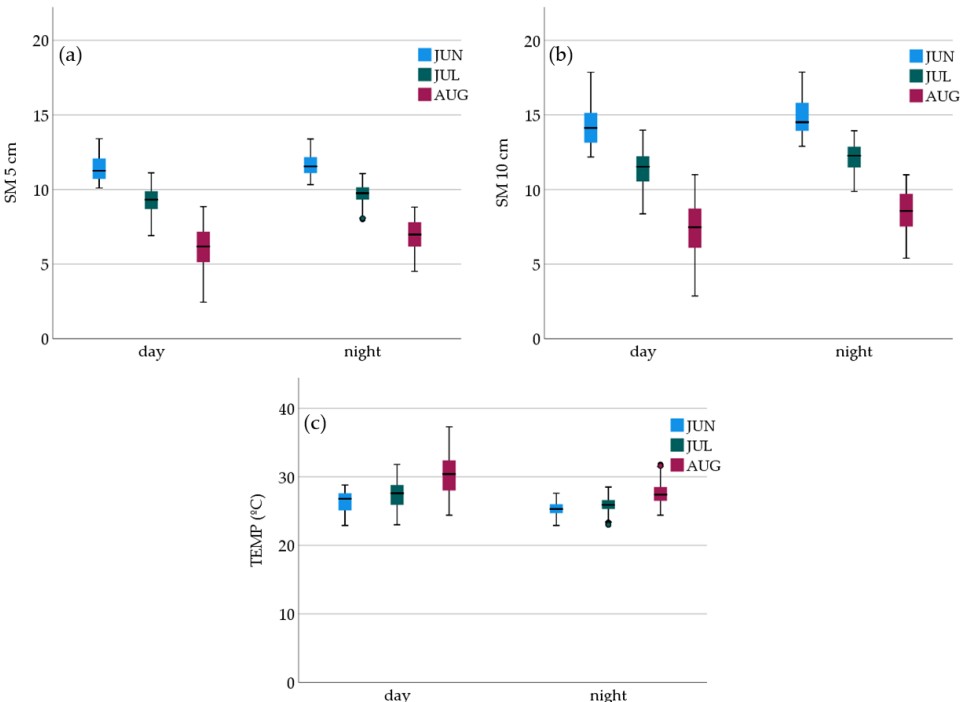

**Figure 8.** Daily variability of soil moisture (SM) and soil temperature by month and time of day. (**a**) SM at 5 cm, (**b**) SM at 10 cm, (**c**) soil temperature.

In August, the temperature continued to increase, a characteristic of the region—the air is drier and there is a greater availability of energy in the form of sensible heat, with an estimated 250 h of sun per month. However, even with higher temperatures during the day, the increase in soil moisture at night continued to occur when the temperature declined (Figure 7). The difference between the air temperature and the soil temperature increased in time (Figure 9); at the end of the wet season in May, the deciduous plants dropped their leaves and the soil was no longer protected from the direct solar radiation. The median values of the soil and air temperature were different in June and August, and the same in July ($p < 0.01$), according to the Wilcoxon test.

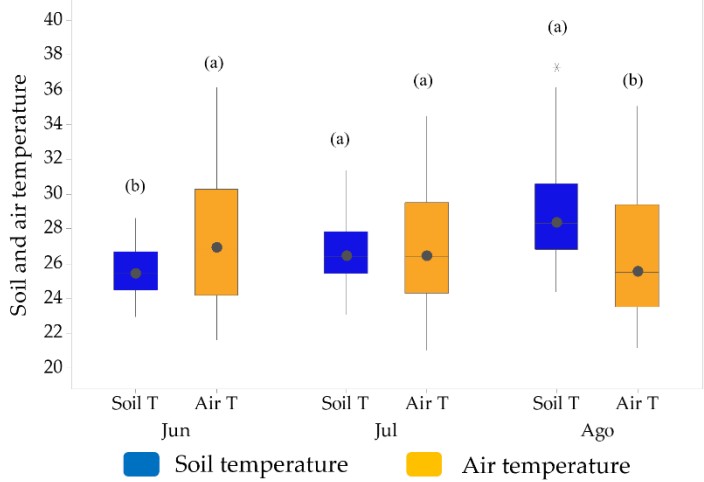

**Figure 9.** Soil and air temperature variability in the period of study. Different letters (a, b) represent a statistically different median at the level of 1% by the Wilcoxon test.

The coefficients of variation values were low (CV < 10%) in June and July, as the variability of the soil moisture is inversely proportional to the water content in the soil [33,34]. This fact was confirmed when we observed the CV values in August (drier soil), where the

CV was more than twice that in the months of June and July (wetter soil). The night period presented a lower CV at both depths throughout the studied period. We believe that the highest CVs during the day can be explained by the higher temperatures recorded during the day, which favor the advection process and consequent moisture losses. In the evening, with the temperature reduction, dew formation and water adsorption favor the addition of water to the soil and promote a reduction of dispersion in soil moisture values. The relationship between the soil temperature and soil moisture was also verified by [12,13,35], in a period characterized by low precipitation and high temperature, where these variables showed a negative correlation.

The soil moisture amplitude increased as the dry season evolved, and was greater at the 10 cm than at the 5 cm depth (Figure 10), with values up to 3.54 mm and 4.52 mm for the depths of 5 cm and 10 cm during the day, and 1.68 mm and 2.18 mm during the night, respectively. Even though the soil moisture content was always higher at 10 cm than at 5 cm, the values tended to converge as the dry season persisted. Towards the end of the dry season, the soil moisture values at both depths were approximately the same (Figure 10), suggesting that the lower layer (10 cm) lost water to the upper layer, as the upper layer lost water to the atmosphere, seeking an equilibrium.

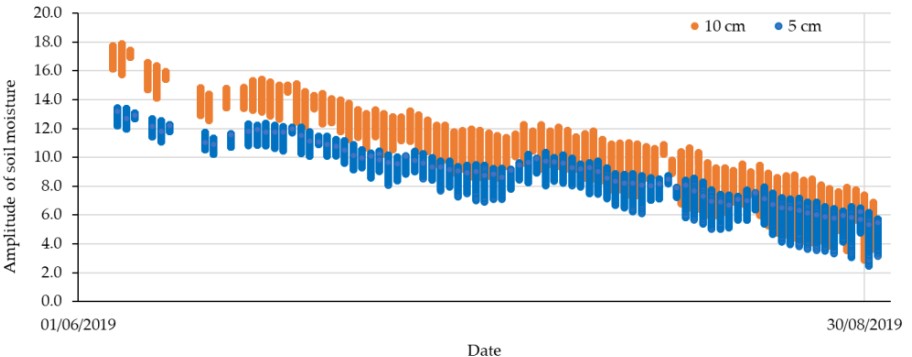

**Figure 10.** Daily amplitude of soil moisture content at 5 cm and 10 cm depths.

The soil moisture responded more to soil temperature at the deeper layer than at the upper layer (Figure 11). This may be because there is more soil moisture at the deeper layer and the water acts as a thermal regulator, emphasizing the sensitivity of the soil moisture to temperature variation (Figure 11).

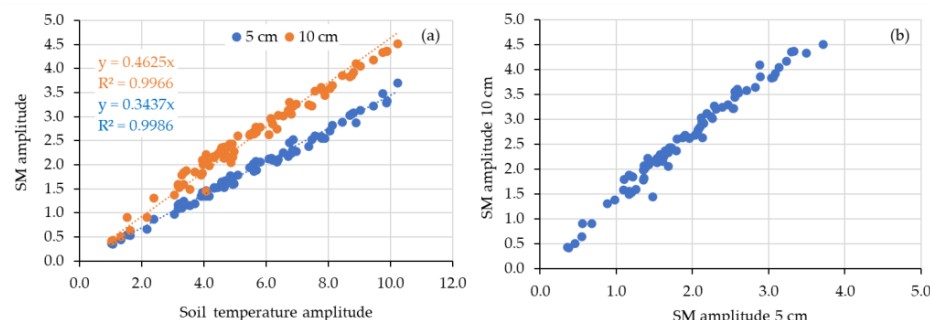

**Figure 11.** (**a**) Soil moisture amplitude vs. soil temperature amplitude; (**b**) soil moisture amplitude at 5 cm vs. soil moisture amplitude at 10 cm.

In June, the day/night variation of the soil moisture was 0.55% and 0.97%, representing 0.28 and 0.49 mm day$^{-1}$, for the depths of 5 and 10 cm, respectively. This is an average daily contribution of non-precipitation water to the soil of 0.28 and 0.49 mm day$^{-1}$. Even though these values represent a total accumulated monthly soil moisture of 8.4 and 14.7 mm during the night, respectively, for depths of 5 and 10 cm, alleviating vegetation water stress, this water is lost during the day. These results were similar to those found by [18] in temperate

dry regions. It is believed that these moisture peaks during the night period are due to the formation of dew. It is known that the condensation of air water vapor at night occurs by reducing the air temperature and contact surface, increasing the relative humidity of the air by cooling [30,36].

During the period of study, the higher average soil moisture content happened during the night when the temperature decreased; the relative humidity of the air increased due to the temperature decrease, and condensation may have occurred, inducing a higher soil moisture content. This increased availability of water ranged from 0.10 to 3.54% and from 0.10 to 4.33% for the depths of 5 and 10 cm, from June to August, respectively, by the adsorption of the water vapor from the soil atmosphere by diffusion. These processes reduce the loss of soil moisture by adding soil moisture to the soil during the night and allow the maintenance of vegetation for longer periods without rainfall. The results presented here support the hypothesis that non-rainfall water input can positively contribute to soil moisture, as was observed by [12,13,15,18]. In dry regions of China, dew formation increased the soil water content between 0.001 and 0.38 mm day$^{-1}$ at a depth of 5 cm [16]. Although these water increases are relatively small, they can contribute to the survival of young plants and account for up to 19% of the annual water intake of semi-arid ecosystems [31].

## 4. Conclusions

Even though the air temperature did not drop below the dew point, there was an increase in soil moisture at night. There was a daily cycle of increase in the soil moisture content at night that was lost during the day, to be gained again the following night. The nightly increase of soil moisture was not enough to maintain the soil moisture content during the dry season, as there was a net loss of soil moisture content up to a depth of 10 cm. The mean daily variation of the gains during the study period was 0.40 and 0.64 mm, for the depths of 5 and 10 cm, respectively—0.28, 0.30 and 0.63 mm/day for the depth of 5 cm, and 0.49, 0.55 and 0.88 mm/day for the depth of 10 cm, in June, July and August, respectively. There is a definite contribution of these nightly gains to alleviate vegetation water stress during the dry months of a dry tropical forest environment. Although three months of experimental data may not be enough to generalize the obtained results, they hep to develop a clearer idea of how crucial the studied process is in reducing the water stress on plants under non-rainfall water input.

**Supplementary Materials:** The following supporting information can be downloaded at: https://www.mdpi.com/article/10.3390/hydrology9060102/s1. Table S1. Detailed data of soil moisture and soil temperature.

**Author Contributions:** Conceptualization, M.S.G. and E.M.d.A.; methodology, H.A.d.Q.P., E.M.d.A. and M.S.G.; formal analysis, M.M.M.d.S., J.B.B. and J.C.R.F.; investigation, E.M.d.A. and M.S.G.; resources, H.A.d.Q.P.; data curation, H.A.d.Q.P.; writing—original draft preparation, M.S.G. and E.M.d.A.; writing—review and editing, M.S.G. and E.M.d.A.; supervision, E.M.d.A.; project administration, E.M.d.A.; funding acquisition, E.M.d.A. and H.A.d.Q.P. All authors have read and agreed to the published version of the manuscript.

**Funding:** This work was supported by the CNPq—Conselho Nacional de Desenvolvimento Científico e Tecnológico, Brazil (grant number 558135/2009-9).

**Institutional Review Board Statement:** Not applicable.

**Informed Consent Statement:** Not applicable.

**Data Availability Statement:** Data is available in Supplementary Materials.

**Acknowledgments:** This study was carried out with the support of the Coordenação de Aperfeiçoamento de Pessoal de Nível Superior-Brasil (CAPES), the Conselho Nacional de Desenvolvimento Científico e Tecnológico (CNPq) and the Fundação Cearense de Apoio ao Desenvolvimento Científico e Tecnológico.

**Conflicts of Interest:** The authors declare that they have no conflict of interest.

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
