# Peer review of "Contribution of Non-Rainfall Water Input to Surface Soil Moisture in a Tropical Dry Forest"

_hydrology, doi:10.3390/hydrology9060102_

Round 1

Reviewer 1 Report

A nicely constructed paper and study.  I think that the results are useful and should be available to researchers.

My comments are all in the attached pdf.

In general all the axes and annotations are too small and hard to read. 

Author Response

Thank you so much for the time you took reviewing our manuscript. It has greatly improved. 
We have accepted all suggestions and recommendations that were available in the pdf. 

As for your questions, we addressed them by including and modifying the text and are presented below: 

Q: Line 55
It is not the same thing. The 200 mm is the total annual precipitation and the 38 mm is the increased soil water content between 0.001 and 0.38-mm day-1 at a depth of 5 cm due to dew formation 

Q: Line 123
The calibration was developed in the study

Q: Line 132
We have rewritten the sentence to address the question: "Soil moisture content and soil temperature data were not normally distributed (p≤0.05) by the Anderson-Darling test. We applied the Wilcoxon test to verify the differences in medians from day to night soil moisture values (p<0.01). "

Q: Line 154

We have rewritten the sentence to address the question: "As this study is focused on the non-rainfall water input, we selected the records from the last rainfall event on June 18th to August 31st (Figure 5), because dew may occur up to three months after the end of the rainy season [22] and may persist throughout the dry season [23]."

Q: Line 164
We have changed the text to pore spaces. It is much better now. Thank you.

Q: Table 2
we have added text that also addresses this phenomena in the Results and Discussion section. "The difference between the air temperature and the soil temperature increased in time (Figure 9) - at the end of the wet season in May, the deciduous plants drop their leaves and the soil is no longer protected to the direct solar radiation."

Q: Line 181
The trend lines show net loss, and we address it in the text: "As the dry season progressed, there was an increase in net losses with some exceptions. "

Q: Line 187
We observed that total radiation from the previous day matters

Q: Table 3
By including radiation in the figure, it gets too confusing. We have opted to add the dates of the days with gains to Figure 7, as the radiation is proportional to the temperature

Q: Line 214
We have already addressed this question previously, please refer to answer to Q: Table 2

Q: Line 224
This unit refers to the depth of water equivalent to the soil moisture content within the analysed depth

Q: Line 237
We have added text to address the question: "At the deeper layer (10 cm), part of the water vapor is transferred and redistributed to the upper layer (5 cm) by diffusion, of which, some is lost to the atmosphere. The hydraulic lift process is not likely to occurr with shallow root depths (Bayala and Prieto, 2020), as is the case of this study area (60 cm), as showed by Pinheiro et al. (2013)."

Q: Line 252
Please refer to answer to Q: Line 237

Q: Line 266
These figures are valuable for agricultural water management 

Reviewer 2 Report

General Comment

The manuscript "Contribution of Non-Rainfall Water Input to Surface Soil Moisture in a Dry Tropical Forest" evaluates the effect of moisture sources non-related to precipitation on the ground. The topic of this manuscript is within this journal's scope and extremely important to understand the vital ecological processes of Tropical Dry Forests (TDFs) strongly linked to water availability. As a forest hydrologist with ample experience in the tropics, I value the authors' effort and the scientific questions behind this work. However, the quality of the current manuscript requires a lot of improvement before its publication in this journal. Therefore, I encourage the authors to improve the experimental design, data analysis, physical evidence, and manuscript writing for further submission.

Major comments:

  1. The introduction requires a significant effort to provide the context on the importance of TDFs (i), the importance of the hydrological processes under dry climate conditions (ii), how these processes affect the vegetation (iii), and the relevance of the research site (iv).
  2. The manuscript underlines the importance of dew as the primary water source describing the changes in soil water moisture (SWM) recorded beneath the forest. However, the experimental design largely omits different processes that may also mislead the authors that soil dew is the primary driver (e.g., dewfall from overstory and understory canopies).
  3. Do the authors consider plant hydraulic lift a tree forest strategy to have water?
  4. Do the authors evaluate the atmospheric air temperature and dew point as a source of water? For example, on cold nights, a large amount of water can condense in the forest canopy and provide a vital water flux by dripping.
  5. Do the authors provide a field calibration for the sensors? This point is crucial because the field experiments usually omit the calibration procedure with soil samples. Also, assuming that all sensors work perfectly from the package may induce wrong results if those are not calibrated to the soil conditions.
  6. May the sensor dependency on temperature be more significant than the soil capacity to produce enough dew to change the SWM signal (Mass Balance Principle)?
  7. The experimental design is not straightforward. Some questions about the placement angle of the sensors, number of sensors, location within the forest, and closeness to plants are critical points (among others) for the type of analysis the authors are carrying out. These variables may strongly influence the final result and lead to wrong conclusions.
  8. The quality of graphs is poor and not standardized. I recommend that the authors homogenize all the plots in terms of terminology, colours, symbols, dates and time formats, units, etc.
  9. Is it available data from a lysimeter that may help to confirm the conclusion of water gains?
  10. How is the evaporation process from the lower soil layer happening to increase the SWM in the upper layer?
  11. What is the vertical soil structure (e.g., soil layers, pit information, soil density at different depths)?
  12. What is the temperature difference between the soil profile and air mass above the soil?
  13. Why the authors are not using a mobile average in the analysis (e.g., Figures 6 and 7)?
  14. The captions of all figures and tables are not informative enough. Also, there is a lack of uniformity in how the units are expressed across the manuscript (e.g., Figure 9 has units missing on the y-axis, the caption Figure 8 shows temperature units as C and not as oC).
  15. The map is not informative enough. It can be more useful if the authors include a diagram of the experimental setup distribution.
  16. The authors use capital letters randomly, figures and tables.
  17. The conclusions are not supported by the analysis, results, or discussion.

Author Response

Thank you so much for the time you took reviewing our manuscript. It has greatly improved, and we hope to have addressed all your questions.

1. The introduction requires a significant effort to provide the context on the importance of TDFs (i), the importance of the hydrological processes under dry climate conditions (ii), how these processes affect the vegetation (iii), and the relevance of the research site (iv).

We have added text to highlight the importance of TDFs: "]. Tropical dry forests (TDF) are present in the five continents, of which 54% is located in South America (Miles et al., 2006), and account for approximately 40% of all tropical forests (Murphy and Lugo, 1986; Miles et al. 2006). These forests are home to the poorest people in the world (CIFOR, 2014)."

We have added text to highlight the importance of the hydrological processes under dry climate conditions: "TDFs show average monthly temperatures above 18 °C, annual precipitation below the 1800 mm isohyet, and rainfall events concentrated within four to seven months (Andrade et al., 2016; Guerreiro et al., 2013). Rainfall is the main water intake in TDFs. The hydrologic regime in these regions depends on the spatial and temporal concentration of rainfall, not being uncommon that 70% of the annual precipitation occurs in only one month [novo 1] or that 90 % of total annual precipitation is registered during the 3-4 months wet season (Andrade et. al, 2020).

We have added text to highlight how these processes affect the vegetation: "These regions are characterized by long periods of water deficit (Guerreiro et al., 2021), and vegetation adapts to the irregular rainfall and extended droughts with small-leaved, thorny trees with twisted trunks, and succulents and therophitic herbs that efficiently respond to the minimal levels of precipitation (Silva et al., 2017). "

The relevance of the research site is addressed in Materials and Methods - Study area, where we locate the study area at the heart of the TDF in northeast Brazil

2. The manuscript underlines the importance of dew as the primary water source describing the changes in soil water moisture (SWM) recorded beneath the forest. However, the experimental design largely omits different processes that may also mislead the authors that soil dew is the primary driver (e.g., dewfall from overstory and understory canopies)

We thank you very much for your comment, as we have addressed the dewfall from overstory and understory canopies simply as dew. We have included text to introduce those processes: "The contribution of non-rainfall water input to the soil is fog, dew, redistribution of condensation from leaves (dewfall from from overstory and understory canopies), hydraulic lift from the deeper to the shallower soil layers, and water vapor adsorption)"

3. Do the authors consider plant hydraulic lift a tree forest strategy to have water?

We have included the hydraulic lift in the introduction, because it may be a water input. However, in this region, the shallow root system does not support the necessary conditions for this process (Bayala and Prieto, 2020), and the study area has a shallow root system (60 cm)- Pinheiro et al., (2013). We have added text in the Results and Discussion section: "The hydraulic lift process is not likely to occurr with shallow root depths (Bayala and Prieto, 2020), as is the case of this study area (60 cm), as showed by Pinheiro et al. (2013)."

4. Do the authors evaluate the atmospheric air temperature and dew point as a source of water? For example, on cold nights, a large amount of water can condense in the forest canopy and provide a vital water flux by dripping. 

We have added text in the Results and Discussion Section to address this question: "During the study period, there were no occurrences of temperatures being below the dew point at any time, therefore, no dew was formed in the atmosphere (https://portal.inmet.gov.br/) (...)"

5. Do the authors provide a field calibration for the sensors? This point is crucial because the field experiments usually omit the calibration procedure with soil samples. Also, assuming that all sensors work perfectly from the package may induce wrong results if those are not calibrated to the soil conditions.

We have a field calibration for the sensors, and we have information on that calibration (also as a function of temperature) in the Materials and Methods section: "From the initially installed four soil moisture sensors, two showed inconsistencies associated to high temperatures due to the electrical characteristics of the soil [18] and data from these sensors was discarded. To minimize the errors due to the high temperatures in the region, the remnant sensors were calibrated for gravimetric soil moisture content by the following equation: 
=12.7+0.756×-0.49×T
where  is the gravimetric soil moisture content (g/g) determined in laboratory,  is the volumetric soil moisture content obtained from the sensor (m3/m3), and T is soil temperature measured at 10 cm depth (C)."

6. May the sensor dependency on temperature be more significant than the soil capacity to produce enough dew to change the SWM signal (Mass Balance Principle)?

We cannot answer this question at this time. Sorry.

7. The experimental design is not straightforward. Some questions about the placement angle of the sensors, number of sensors, location within the forest, and closeness to plants are critical points (among others) for the type of analysis the authors are carrying out. These variables may strongly influence the final result and lead to wrong conclusions.

We have added text in the Materials and Methods section to address this question: "From the initially installed four soil moisture sensors, perpendicular to the soil surface and under the natural vegetation, (...)"

8. The quality of graphs is poor and not standardized. I recommend that the authors homogenize all the plots in terms of terminology, colours, symbols, dates and time formats, units, etc.

We have improved the quality of the graphs. Hope it is much better now. 

9. Is it available data from a lysimeter that may help to confirm the conclusion of water gains?

There is no lysimeter in the area to help confirm the conclusion of the water gains. 

10. How is the evaporation process from the lower soil layer happening to increase the SWM in the upper layer?

We have added text in the Results and Discussion Section to address this question: "As the soil temperature rises, adsorbed water in the soil evaporates to the soil air medium and is redistributed to the pore spaces. At the deeper layer (10 cm), part of the water vapor is redistributed to the upper layer (5 cm) by diffusion (...)"

11. What is the vertical soil structure (e.g., soil layers, pit information, soil density at different depths)?

We have added text in the Material and Methods section to address this question: "The soil of the area is classified as a typical Carbonate Ebanic Vertisol [15], belongs to a sedimentary rocks formation with limestone rock below 36 cm, and the presence of A, B and C horizons at 0-5 cm, 5-36 cm, and above, respectively. "

12. What is the temperature difference between the soil profile and air mass above the soil?

We have added text in the Results and Discussion Section and a Figure to address this question: "The difference between the air temperature and the soil temperature increased in time (Figure 9) - at the end of the wet season in May, the deciduous plants drop their leaves and the soil is no longer protected to the direct solar radiation."

13. Why the authors are not using a mobile average in the analysis (e.g., Figures 6 and 7)?

We did not want to see the tendency, but rather the daily variation in order to see if the previous daily characteristics were significant for the differences observed in the day. 

14. The captions of all figures and tables are not informative enough. Also, there is a lack of uniformity in how the units are expressed across the manuscript (e.g., Figure 9 has units missing on the y-axis, the caption Figure 8 shows temperature units as C and not as C).

15. The map is not informative enough. It can be more useful if the authors include a diagram of the experimental setup distribution.

We have included more information in Figure 1

16. The authors use capital letters randomly, figures and tables. 

We did not understand your comment, but have searched for inconsistencies

17. The conclusions are not supported by the analysis, results, or discussion.

We have rearranged the text and added to the conclusions. We have added text in the Results and Discussion section to address this question (e.g., 4., 10 and 12.)

Reviewer 3 Report

Dear Authors,

please find my comments on your manuscript.

Regards

Reviewer

Author Response

Thank you so much for the time you took reviewing our manuscript. It has greatly improved. 

1. Introduction
line 59-63: please write more on the other studies concerning this topic
We have added multiple references and text to address this suggestion

2. Material and Methods
Study area and all other subsections must be enumerated.
It has been done

Figure 1: the quality of the graph is very low. Coordinates can be bearly read. Indication of the study area is also very poorly marked on the map.
It has been corrected

Table 1: please provide references. Do those values come from measurements performed by the authors?
The analysis was performed by the authors using standard procedures (referenced in the text).

Figure 3: must be seriously reorganized. You must mark which part is A and which one is B. Also, the bottom figure is too small to be read. The same type of comment refers to Figure 4 as well.
It has been corrected

line 124: all equations must be written using mathematical expressions according to the journal style. What are the arguments behind the application of this equation in your study? 
It has been corrected. The equation addresses the fact that the sensor was calibrated using laboratory determined soil moisture by the gravimetric method

3. Results and Discussion
Figure 5: acronyms SM must be explained on the graph or figure caption.
It has been done

Figure 10: please reorganize it as in the current version. It looks ugly to compare two plots of different sizes.
we have reorganized it. Hope it is more attractive now

4. Conclusions
This part is too brief and must be expanded. Also, please provide a summary part
We have rearranged the text and added to the conclusions. Although we acknowledge that some journals suggest a summary in the conclusions, we believe that that should be done in the abstract only.

Reviewer 4 Report

The paper  Contribution of Non-Rainfall Water Input to Surface Soil  Moisture in a Dry tropical Forest, brings interesting results from experimental measurements which helped to evaluate dew contribution to soil moisture content at the soil surface in minimizing the water stress on plants with predominant surface roots.

The results of this study are based on the experimental measurement in one study site, which was performed from Jun/2019 to 27 Aug/2019.

I have some comments which should help to improve the presented results.

The Author's contribution to the present-day level of knowledge may be in better explaining several parts of the study.

For E.g. the introduction needs a more detailed analysis of the previous results dealing with the topic of this study, not only adding the citations. The mentioned results should then be discussed with the author's findings in the discussion part of the study, which is missing.

Also, the research question of this study needs to be pointed out.

Some Figs are not good readable, and the Conclusions are too short.

As a weak point of this study, I find that the time of the experimental measurements is only two months, which is too short to make serious conclusions. Furthermore, the presented results are time-based; the comparison to other studies or measurements from other places in the same climatic zone is missing.

One possibility to improve the study is to compare the results with other data, e.g. data from satellite measurements. For example, satellite soil moisture products are available in daily time step for various soil depths. It would be very interesting to compare this data with the measured ones in this study and look at history, and the future, how the SM data are changing. This approach would be of interest also for practical application in water resources management in the region.

Author Response

Thank you so much for the time you took reviewing our manuscript. It has greatly improved. 

Q: For E.g. the introduction needs a more detailed analysis of the previous results dealing with the topic of this study, not only adding the citations. The mentioned results should then be discussed with the author's findings in the discussion part of the study, which is missing.

We have included new text to address your question, and have supported our findings with studies developed by other scientists. 

Q: Also, the research question of this study needs to be pointed out.
The research question was to determine the non-rainfall water contribution to soil moisture in a tropical dry forest with shallow roots vegetation, and it has been addressed in the text. Nonetheless, we have added the following text to better describe our hypothesis: "In the actual context of climate changes, it is important to assess more water sources in TDS as the main rainfall input is expected to decrease and their contribution to the system. The objective of the study was to evaluate non-rainfall water contribution to soil moisture content at the soil surface and how it minimizes the water stress on plants with predominantly surface roots."

Q: Some Figs are not good readable, and the Conclusions are too short.
We have worked on the figures to be more readable. Thank you for the suggestion. As for the conclusions, we have rearranged the text and added to the conclusions.

Q: As a weak point of this study, I find that the time of the experimental measurements is only two months, which is too short to make serious conclusions. Furthermore, the presented results are time-based; the comparison to other studies or measurements from other places in the same climatic zone is missing.
We have acquired data for the whole year, but we wanted to focus on the dry period, when there is not rainfall. Even though there were "only" three months of presented data, they were collected at one-minute intervals, which totals above 390,000 entries (130,000 for soil moisture at two depths and for temperature). During the three months under study, there was no rainfall and, therefore, we wanted to evaluate and explain the source for the increase in soil moisture content, for it will have a positive impact in keeping the vegetation through the harsh dry period.
We have added text to both the introduction and results and discussion sections to improve and better support the analysis. Thank you for the reminder. We have added text to compared the obtained results with other studies: e.g., "The relationship between soil temperature and soil moisture was also verified by Feng [26], in a period characterized by low precipitation and high temperature, where these variables showed a negative correlation."; "In dry regions of China, dew formation increased soil water content between 0.001 and 0.38-mm day-1 at a depth of 5 cm [10]. "; "These results were similar to those found by [9] in temperate dry regions. It is believed that these moisture peaks during the night period are due to the formation of dew. It is known that the condensation of air water vapor at night occurs by reducing the air temperature and contact surface, increasing the relative humidity of the air by cooling [22,27]."

Q: One possibility to improve the study is to compare the results with other data, e.g. data from satellite measurements. For example, satellite soil moisture products are available in daily time step for various soil depths. It would be very interesting to compare this data with the measured ones in this study and look at history, and the future, how the SM data are changing. This approach would be of interest also for practical application in water resources management in the region.
Your suggestion is very, very interesting, but it was not the subject of our study. The objective of the study was to assess the contribution non-rainfall water at night to soil moisture content at the soil surface and how it reduces the water stress on plants in a region with predominantly surface roots, and not to validate the soil moisture content. 

Round 2

Reviewer 3 Report

Dear Authors,

I am happy with the changes that you have incorporated into the manuscript. In my opinion, it has been significantly improved and is now ready for being published. Thanks, for this interesting study.

Kind regards

Reviewer

Author Response

Thank you very much for your time in reviewing our manuscript. It has greatly improved our work. 

Reviewer 4 Report

-

Author Response

Thank you so much for your time in reviewing our manuscript. It has greatly improved it.